# Extensive ribosome and RF2 rearrangements during translation termination

Egor Svidritskiy[1], Gabriel Demo[1], Anna B Loveland[1], Chen Xu[2],
Andrei A Korostelev[1,2]*

[1]RNA Therapeutics Institute, University of Massachusetts Medical School,
Worcester, United States; [2]Biochemistry and Molecular Pharmacology, University of
Massachusetts Medical School, Worcester, United States

**Abstract** Protein synthesis ends when a ribosome reaches an mRNA stop codon. Release factors
(RFs) decode the stop codon, hydrolyze peptidyl-tRNA to release the nascent protein, and then
dissociate to allow ribosome recycling. To visualize termination by RF2, we resolved a cryo-EM
ensemble of *E. coli* 70S•RF2 structures at up to 3.3 Å in a single sample. Five structures suggest a
highly dynamic termination pathway. Upon peptidyl-tRNA hydrolysis, the CCA end of deacyl-tRNA
departs from the peptidyl transferase center. The catalytic GGQ loop of RF2 is rearranged into a
long β-hairpin that plugs the peptide tunnel, biasing a nascent protein toward the ribosome exit.
Ribosomal intersubunit rotation destabilizes the catalytic RF2 domain on the 50S subunit and
disassembles the central intersubunit bridge B2a, resulting in RF2 departure. Our structures
visualize how local rearrangements and spontaneous inter-subunit rotation poise the newly-made
protein and RF2 to dissociate in preparation for ribosome recycling.
DOI: https://doi.org/10.7554/eLife.46850.001

*For correspondence:
andrei.korostelev@umassmed.edu

Competing interests: The
authors declare that no
competing interests exist.

Reviewing editor: Sjors HW
Scheres, MRC Laboratory of
Molecular Biology, United
Kingdom

## Introduction

Ribosomes terminate translation when release factors (RFs) recognize an mRNA stop codon in the A
(aminoacyl-tRNA) site (*Brenner et al., 1967*; *Brenner et al., 1965*). Upon stop-codon recognition,
the release factor hydrolyzes peptidyl-tRNA in the P site of the large subunit, thus releasing nascent
protein from the ribosome (*Capecchi, 1967a*; *Capecchi, 1967b*; *Scolnick et al., 1968*;
*Tompkins et al., 1970*). In bacteria, RF1 recognizes UAA or UAG, and RF2 recognizes UAA or UGA
codons (*Nakamura et al., 1995*). Following peptide release, RF1 or RF2 must dissociate from the
ribosome to allow the ribosome and release factor to recycle. While structural studies have provided
snapshots of some peptidyl-tRNA hydrolysis steps (reviewed in *Dunkle and Cate, 2010*; *Koroste-
lev, 2011*; *Ramakrishnan, 2011*; *Rodnina, 2018*; *Youngman et al., 2008*), recent biophysical and
biochemical findings suggest a highly dynamic series of termination events (*Adio et al., 2018*;
*Indrisiunaite et al., 2015*; *Prabhakar et al., 2017*; *Sternberg et al., 2009*), which have not been
structurally visualized.

In the pre-termination (pre-hydrolysis) state, the substrate peptidyl-tRNA resides in a non-rotated
(classical) conformation of 70S ribosome, with its anticodon stem loop bound to the mRNA codon in
the P site of the small 30S subunit and the 3′-terminal CCA-peptidyl moiety in the P site of the large
50S subunit (*Agrawal et al., 2000*; *Ermolenko et al., 2007*). Cryogenic electron microscopy (cryo-
EM) and X-ray studies reveal RF1 or RF2 bound in the A site next to P tRNA (*Klaholz et al., 2003*;
*Petry et al., 2005*; *Rawat et al., 2006*; *Rawat et al., 2003*), and provide insight into pre-hydrolysis
(*Jin et al., 2010*) and post-hydrolysis states (*Korostelev et al., 2008*; *Korostelev et al., 2010*;
*Laurberg et al., 2008*; *Weixlbaumer et al., 2008*). The release factor's codon-recognition domain

(domain 2) binds the stop codon in the 30S decoding center. The catalytic GGQ motif (glycine-gly-cine-glutamine of domain 3) inserts into the 50S peptidyl-transferase center (PTC) adjacent to the CCA end of P-tRNA. Ribosomes retain the non-rotated conformation in RF-bound structures, including those with mutated or post-translationally modified RFs (*Pierson et al., 2016*; *Santos et al., 2013*; *Zeng and Jin, 2018*) or when RFs are perturbed by the termination inhibitor blasticidin S (*Svidritskiy and Korostelev, 2018a*; *Svidritskiy and Korostelev, 2018b*) or by a sense codon in the A site (*Svidritskiy et al., 2018*). The catalytic GGQ loop adopts the same compact conformation comprising a short α-helix in pre-hydrolysis-like and post-hydrolysis structures, as discussed below.

Ribosome and RF dynamics are implicated in post-hydrolysis steps. In *E. coli*, RF1 dissociation is assisted by the non-essential GTPase RF3 (*Freistroffer et al., 1997*; *Koutmou et al., 2014*), whereas RF2 can spontaneously dissociate from the ribosome (*Adio et al., 2018*). RF3 induces rotation of the small subunit relative to the large subunit by ~9 degrees in the absence of RF1 and RF2 (*Ermolenko et al., 2007*; *Zhou et al., 2012*) and stabilizes a 'hybrid' state of deacylated tRNA (P/E) (*Gao et al., 2007*; *Jin et al., 2011*). In the P/E conformation, the anticodon stem loop remains in the 30SS P site, while the elbow and acceptor arm bind the L1 stalk and E (exit) site of the 50S subunit. A recent cryo-EM study (*Graf et al., 2018*) showed that if RF1 is locked on the post-hydrolysis ribosome by antibacterial peptide apidaecin 137 (*Florin et al., 2017*), the ribosome can undergo intersubunit rotation in the presence of both RF1 and RF3. By contrast, biochemical experiments showed that locking RF1 on the ribosome by inactivating its catalytic motif (GGQ->GAQ) prevented RF1 recycling by RF3 from the pre-hydrolysis ribosome (*Koutmou et al., 2014*), echoing the deficiency of mutant RF2 (GGQ->GAQ) in stimulating GTP hydrolysis by RF3 (*Zavialov et al., 2002*). These and other (*Adio et al., 2018*) observations indicate that peptidyl-tRNA hydrolysis is critical for RF dissociation, likely due to propensity of deacylated tRNA for exiting the P site and sampling the P/E hybrid state (*Moazed and Noller, 1989*) via spontaneous intersubunit rotation (*Cornish et al., 2008*). Nevertheless, RF3 is dispensable for growth of *Escherichia coli* (*Grentzmann et al., 1994*; *Mikuni et al., 1994*), and its expression is not conserved in bacteria (*Leipe et al., 2002*). For example, RF3 is not present in the thermophilic model organisms of the *Thermus* and *Thermatoga* genera and in infectious *Chlamydiales* and *Spirochaetae*. This means that both RF1 and RF2 are capable of performing a complete round of termination independently of RF3. Indeed, recent Förster Resonance Energy Transfer (FRET) studies demonstrate spontaneous RF1 and RF2 dissociation after peptidyl-tRNA hydrolysis (*Prabhakar et al., 2017*), coupled with transient RF sampling of the rotated ribosomes (*Adio et al., 2018*). The structural basis for RF dissociation, however, remains incompletely understood because RFs have not been structurally visualized on ribosomes with rearranged/rotated subunits in the absence of RF3 and inhibitors.

In this work, we used ensemble cryo-EM, as described earlier (*Abeyrathne et al., 2016*; *Loveland et al., 2017*), employing 3D maximum likelihood classification/sorting of particle images (*Lyumkis et al., 2013*; *Scheres, 2010*; *Scheres et al., 2007*) to visualize the structural dynamics of RF2 on the 70S ribosome (*Figure 1*, *Figure 1—figure supplement 1*, *Figure 1—figure supplement 2* and *Figure 1—figure supplement 3*). The structures capture RF2 and tRNA in distinct 70S conformational states (*Table 1*). They reveal an unexpected rearrangement of the catalytic GGQ loop, which suggests that RF2 helps direct nascent protein out of the ribosome. Our findings reconcile biophysical observations and, together with other structural studies, allow a reconstruction of the termination course by RF2, from initial binding to dissociation.

## Results and discussion

We formed a termination complex using *E. coli* 70S ribosomes and RF2 to catalyze the hydrolysis of substrate N-formyl-methionyl-tRNA$^{fMet}$ (fMet-tRNA$^{fMet}$) bound to mRNA carrying an AUG codon in the P site and a UGA stop codon in the A site. Although the stop codon is placed immediately after the start codon, unlike in cellular mRNAs, such model termination complexes have been successfully used in numerous structural, biochemical and biophysical studies (*Adio et al., 2018*; *Casy et al., 2018*; *Koutmou et al., 2014*; *Kuhlenkoetter et al., 2011*; *Laurberg et al., 2008*; *Pallesen et al., 2013*; *Pierson et al., 2016*; *Shi and Joseph, 2016*; *Sternberg et al., 2009*). Indeed, under the conditions used for cryo-EM sample preparation in this work, RF2 supports hydrolysis of fMet-tRNA$^{fMet}$ and release of fMet (*Figure 1—figure supplement 4*), demonstrating activity of the ribosome and RF2 in our sample. Maximum-likelihood classification of cryo-EM data in FREALIGN (*Lyumkis et al.,*

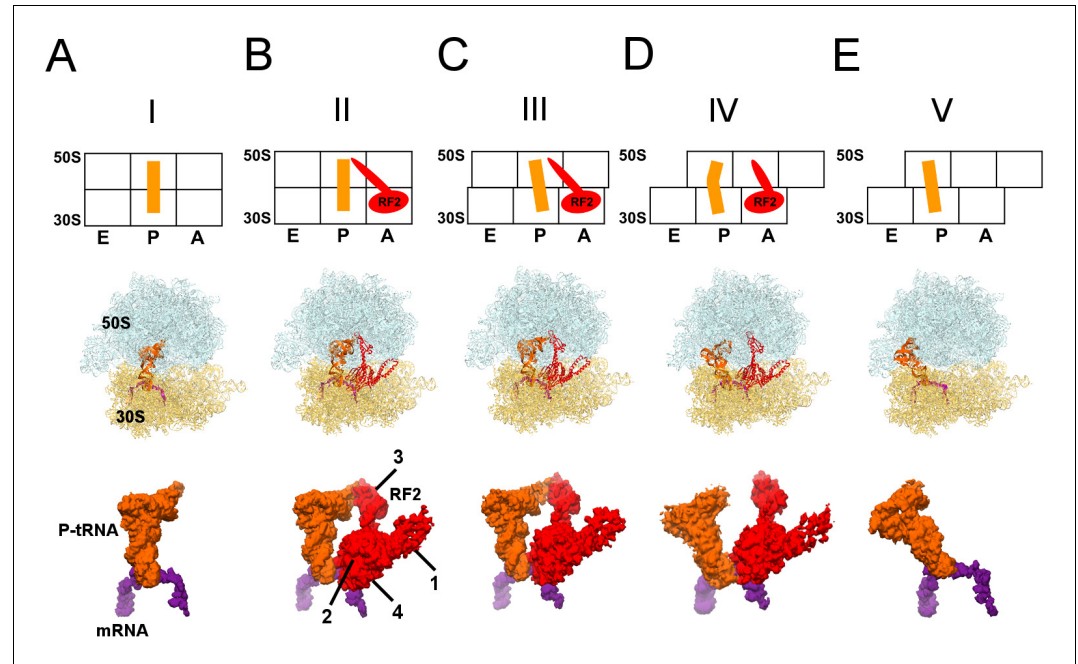

**Figure 1.** Cryo-EM Structures I to V. Panels A-E show for each structure: *upper row:* a schematic of the conformations of the 70S ribosome, tRNA and RF2, with ribosomal subunits (50S and 30S) and A, P and E sites labeled; *middle row:* structures with RF2 shown in red, tRNA$^{fMet}$ in orange, 30S subunit in yellow, 50S subunit in cyan and mRNA in purple; *lower row:* cryo-EM density for mRNA, tRNA and RF2, colored as in the middle row. Domains of RF2 are labeled in panel B, illustrating the codon-recognition superdomain (domains 2 and 4) and catalytic domain (3).

DOI: https://doi.org/10.7554/eLife.46850.002

The following figure supplements are available for figure 1:

**Figure supplement 1.** Maximum-likelihood classification strategies to obtain the final maps and state occupancies (shown as percentages of particles for each classification).
DOI: https://doi.org/10.7554/eLife.46850.003

**Figure supplement 2.** Global and local resolution of Structures I-V.
DOI: https://doi.org/10.7554/eLife.46850.004

**Figure supplement 3.** A representative micrograph of the 70S•RF2 sample showing ribosome particles.
DOI: https://doi.org/10.7554/eLife.46850.005

**Figure supplement 4.** RF2 mediates formyl-methionine release from 70S ribosomes.
DOI: https://doi.org/10.7554/eLife.46850.006

*2013*) yielded five structures at average resolutions of 3.3 Å to 4.4 Å (*Table 2*; *Figure 1*). The structures differ by the degrees of intersubunit rotation and 30S head rotation (swivel), and are consistent with the following termination stages: pre-termination (i.e., no RF2; Structure I); peptidyl-tRNA

**Table 1.** Cryo-EM structures obtained from a single sample of the 70S•RF2 complex.

|  | I | II | III | IV | V |
|---|---|---|---|---|---|
| tRNA and RF2 occupancy | P-tRNA | P-tRNA, RF2 | P-like (L5) tRNA, RF2 | P/E-like (L1) tRNA, RF2 | P/E tRNA |
| 30S (body) rotation | 1.6° | 1.6° | 4.9° | 7.4° | 6.8° |
| 30S head rotation (swivel) | 2.0° | 1.4° | 1.7° | 2.8° | 6.0° |
| Resolution (FSC = 0.143) | 3.3 Å | 3.3 Å | 3.7 Å | 4.4 Å | 3.4 Å |

DOI: https://doi.org/10.7554/eLife.46850.007

**Table 2.** Data collection and refinement statistics for cryo-EM Structures I-V.

| STRUCTURE | I | II | III | IV | V |
|---|---|---|---|---|---|
| PDB ID | 6OFX | 6OG7 | 6OGF | 6OGG | 6OGI |
| EMD code | 20048 | 20052 | 20056 | 20057 | 20058 |
| Data collection | | | | | |
| EM equipment | FEI Titan Krios | | | | |
| Voltage (kV) | 300 | | | | |
| Detector | K2 summit | | | | |
| Pixel size (Å) | 1.33 | | | | |
| Electron dose ($e^-/Å^2$) | 29.4 | | | | |
| Defocus range (μm) | $- 0.5 - 1.8$ | | | | |
| Reconstruction | | | | | |
| Software | Frealign v9.11 | | | | |
| Number of particles in final map | 102,723 | 62,029 | 28,549 | 5,881 | 63,383 |
| Final resolution (Å) | 3.3 | 3.3 | 3.7 | 4.4 | 3.4 |
| Average sharpening $B$ factor ($Å^2$) | -30 | -26 | -17 | -4 | -30 |
| Structure Refinement | | | | | |
| Model Fitting | Chimera/Pymol | | | | |
| Refinement | | | | | |
| Software | RSRef/Phenix | | | | |
| Correlation Coefficient,cc_mask [*] | 0.83 | 0.84 | 0.84 | 0.76 | 0.82 |
| Real-space R-factor [†] | 0.25 | 0.23 | 0.22 | 0.25 | 0.25 |
| Validation (proteins) | | | | | |
| MolProbity Score [‡] | 2.22 | 2.34 | 2.42 | 2.26 | 2.24 |
| Clash score, all atoms [‡] | 17.4 | 17.4 | 16.5 | 15.1 | 16.3 |
| Poor rotamers (%) [‡] | 0.4 | 0.7 | 0.7 | 0.6 | 0.5 |
| Favored/allowed rotamers (%) [‡] | 99.6 | 99.3 | 99.3 | 99.4 | 99.5 |
| *Ramachandran-plot statistics* | | | | | |
| Outlier (%) [‡] | 0 | 0.6 | 0.8 | 0.0 | 0.4 |
| Favored (%) [‡] | 92.2 | 88.1 | 82.3 | 89.0 | 90.7 |
| *R.m.s. deviations* [†,§] | | | | | |
| Bond length (Å) | 0.008 | 0.006 | 0.006 | 0.006 | 0.009 |
| Bond angle (°) | 1.064 | 0.892 | 0.922 | 0.848 | 1.113 |
| Validation (RNA) | | | | | |
| Good sugar puckers (%) [‡] | 99.7 | 99.8 | 99.8 | 99.8 | 99.8 |
| Good backbone conformation (%) [‡] | 82.9 | 84.6 | 85.0 | 85.0 | 82.0 |

[*]from Phenix

[†]from RSRef

[§]root-mean-square deviations

[#] RNA backbone suites that fall into recognized rotamer conformations defined by MolProbity

DOI: https://doi.org/10.7554/eLife.46850.008

hydrolysis by RF2 (Structure II); tRNA exit from the PTC (Structure III); destabilization of the catalytic domain of RF2 (Structure IV); and post-termination (i.e., after RF2 dissociates; Structure V).

Pre-termination-like Structure I contains tRNA in the P site with its 3′-CCA end positioned in the peptidyl transferase center (*Figure 1A*). Although the tRNA is likely partially or completely

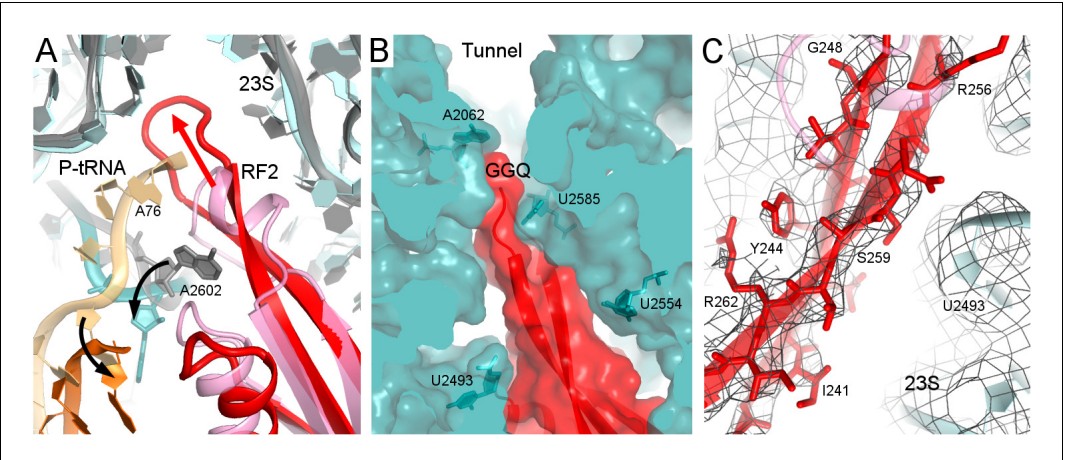

**Figure 2.** Rearrangement of the catalytic domain of RF2 upon tRNA displacement from the PTC. (**A**) Rearrangements of RF2 in Structure II (RF2 in red, P-tRNA in orange and 23S in cyan) in comparison with the canonical 70S•RF2 conformation (X-ray structure: *Korostelev et al., 2008*; RF2 in pink, P-tRNA in light orange, 23S in gray). Arrows show rearrangements for RF2 (red), tRNA acceptor arm and 23S nucleotide A2602 (black). The superposition was achieved by structural alignment of 23S rRNA. (**B**) The GGQ region of RF2 forms a long β-hairpin that reaches into the constriction of the peptide tunnel (at A2062) and thus plugs the tunnel. Nucleotides of 23S rRNA and the GGQ motif are labeled. (**C**) Cryo-EM density for the β-hairpin formed by the GGQ region. Residues of RF2 and 23S rRNA (also shown in panel B for reference) are labeled.

DOI: https://doi.org/10.7554/eLife.46850.009

The following figure supplements are available for figure 2:

**Figure supplement 1.** Rearrangements of PTC nucleotides (cyan) upon formation of the β-hairpin structure by the catalytic region of RF2 (red; Structure II is shown).

DOI: https://doi.org/10.7554/eLife.46850.010

**Figure supplement 2.** Cryo-EM density (mesh) shows that the tip of the GGQ loop of RF2 is poorly ordered (gray model) when the catalytic domain adopts the β-hairpin conformation (red; Structure II is shown).

DOI: https://doi.org/10.7554/eLife.46850.011

---

deacylated due to the action of RF2, the tRNA and ribosome conformations closely resemble those in pre-termination aminoacyl- and peptidyl-tRNA complexes (*Adio et al., 2018*; *Cornish et al., 2008*; *Jin et al., 2010*; *Polikanov et al., 2014*). The 30S subunit adopts a conformation similar to crystal structures of non-rotated ribosomes with or without RFs (*Jin et al., 2010*; *Korostelev et al., 2008*; *Korostelev et al., 2006*; *Korostelev et al., 2010*; *Laurberg et al., 2008*; *Selmer et al., 2006*; *Weixlbaumer et al., 2008*), and exhibits a slight ~1.5° rotation relative to the 70S•RF2 crystal structure formed with the UGA stop codon (*Weixlbaumer et al., 2008*).

RF2 is bound in Structures II to IV (*Figure 1B–D*). The 70S intersubunit rotation ranges from ~1.5° in Structure II to ~7.5° in Structure IV (*Table 1*). This nearly full range of rotation is typical of translocation complexes with tRNA (*Agirrezabala et al., 2008*; *Agirrezabala et al., 2012*; *Dunkle et al., 2011*; *Fischer et al., 2010*; *Julián et al., 2008*) and elongation factor G (EF-G) (*Brilot et al., 2013*; *Chen et al., 2013*; *Frank and Agrawal, 2000*; *Gao et al., 2009*; *Pulk and Cate, 2013*; *Tourigny et al., 2013*; *Zhou et al., 2013*).

The most well-resolved 3.3 Å resolution Structure II (*Figure 1B*) accounts for the largest population of RF2-bound ribosome particles in the sample (*Table 2*), consistent with stabilization of the non-rotated 30S states by RF2 observed in FRET studies (*Casy et al., 2018*; *Prabhakar et al., 2017*; *Sternberg et al., 2009*). The overall extended conformation of RF2 resembles that seen in published 70S•RF2 structures (*Korostelev et al., 2008*; *Weixlbaumer et al., 2008*) with its codon-recognition superdomain (domains 2 and 4; *Figure 1B*) bound to the UGA stop codon in the decoding center of the 30S subunit and its catalytic domain 3 inserted into the peptidyl-transferase center.

Structure II features RF2 with a dramatically rearranged catalytic loop, coinciding with the lack of density for the CCA end of the P tRNA. In previous pre- and post-hydrolysis-like structures, the P-tRNA CCA end binds the PTC, and RF2 forms a compact catalytic loop (residues 245–258). In

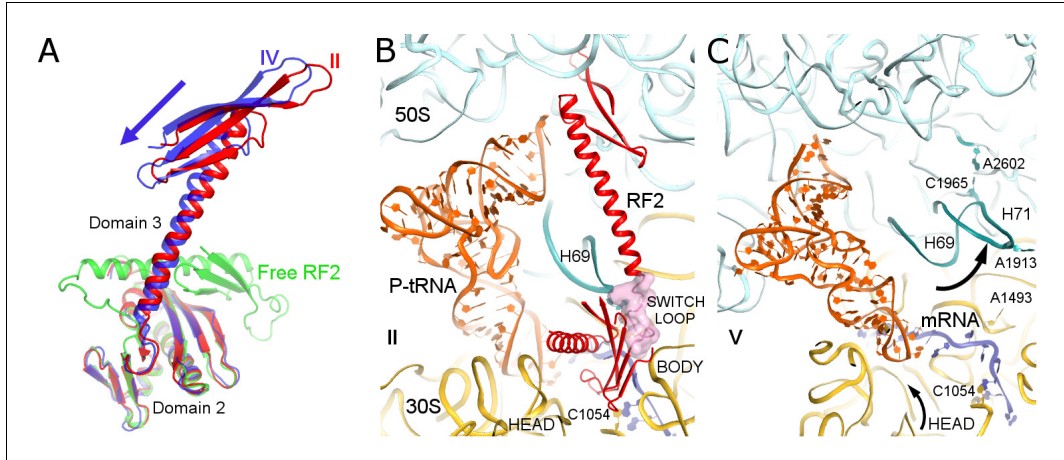

**Figure 3.** Differences in conformations of RF2 and helix 69 of 23S rRNA in Structures II, IV and V. (**A**) RF2 in Structures II (red) and IV (blue) in comparison with the crystal structure of isolated (free) RF2 (green, PDB 1GQE [*Vestergaard et al., 2001*]). The arrow shows the direction of domain three repositioning from Structure II to IV to free RF2. Structures were aligned by superposition of domain 2. (**B**) Interactions of H69 (teal) with the switch loop (pink surface) of RF2 in Structure II (RF2 in red, P-tRNA in orange, 50S in cyan and 30S in yellow). Domains 2 and 3 of RF2, head and body domains of the 30S subunit and nucleotides A1913 (at H69) and C1054 (head) are labeled. (**C**) In Structure V, dissociation of H69 from the decoding center and packing on H71 next to A2602 disassembles intersubunit bridge B2a. The structure is colored as in panel B. View in panels B and C is rotated by ~180° relative to panel **A**.

DOI: https://doi.org/10.7554/eLife.46850.012

The following figure supplement is available for figure 3:

**Figure supplement 1.** Structure alignments show that the disengaged H69 in Structure V (teal) is incompatible with the extended (red, Structure II) and compact (pink, PDB 1GQE [*Vestergaard et al., 2001*]) conformations of RF2 due to steric hindrance.

DOI: https://doi.org/10.7554/eLife.46850.013

these previous structures, the catalytic $^{250}GGQ^{252}$ motif resides at the tip of a short α-helix, adjacent to the terminal tRNA nucleotide A76 (*Figure 2A*, light shades; (*Jin et al., 2010*; *Korostelev et al., 2008*; *Laurberg et al., 2008*; *Weixlbaumer et al., 2008*). The α-helical region is held in place by A2602 of 23S ribosomal RNA (rRNA), which is critical for termination efficiency (*Amort et al., 2007*; *Polacek et al., 2003*). In Structure II, however, the catalytic loop adopts an extended β-hairpin conformation that reaches ~10 Å deeper into the peptide tunnel (*Figure 2A–B*). To accommodate the extended β-hairpin, nucleotides G2505 and U2506 of 23S rRNA are shifted to widen the PTC (*Figure 2—figure supplement 1A*). The center of the β-hairpin occupies the position normally held by A76 of P tRNA (*Figure 2A*). The lack of density for the CCA moiety thus suggests that the 3′ end of P tRNA is at least partially released from the PTC. Indeed, formation of the β-hairpin and release of the CCA moiety is accompanied by a slight shift in the P-tRNA acceptor arm away from the PTC and by ~160° rotation of nucleotide A2602 to form a Hoogsteen base pair with C1965 at helix 71 of 23S rRNA (*Figure 2A* and *Figure 2Figure 2—figure supplement 1B*). The cryo-EM map shows strong density for the extended β strands (*Figure 2C*), but weaker density for the GGQ residues at the tip of the hairpin, consistent with conformational heterogeneity of the flexible glycine backbone (*Figure 2—figure supplement 2*). Nevertheless, the tip of the β-hairpin plugs the narrowest region of the peptide tunnel at A2062 of 23S rRNA (*Figure 2B*). Thus, Structure II appears to represent a previously unseen post-peptide-release state with deacyl-tRNA dissociating from the PTC.

In Structure III (*Figure 1C*), a 5° rotation of the 30S subunit shifts the tRNA elbow by ~10 Å along protein L5 toward the E site. As a result, the P-tRNA acceptor arm pulls further out of the 50S P site, and the end of the acceptor arm helix points at helix α7 of RF2 near positively charged Lys 282 and Lys 289. The CCA moiety remains unresolved. The 5° intersubunit rotation also causes a slight 2° rotation of RF2 domain 3 relative to domain 2, but the overall conformation of RF2 remains similar to that in Structure II, with the tip of the extended β-hairpin plugging the peptide exit tunnel.

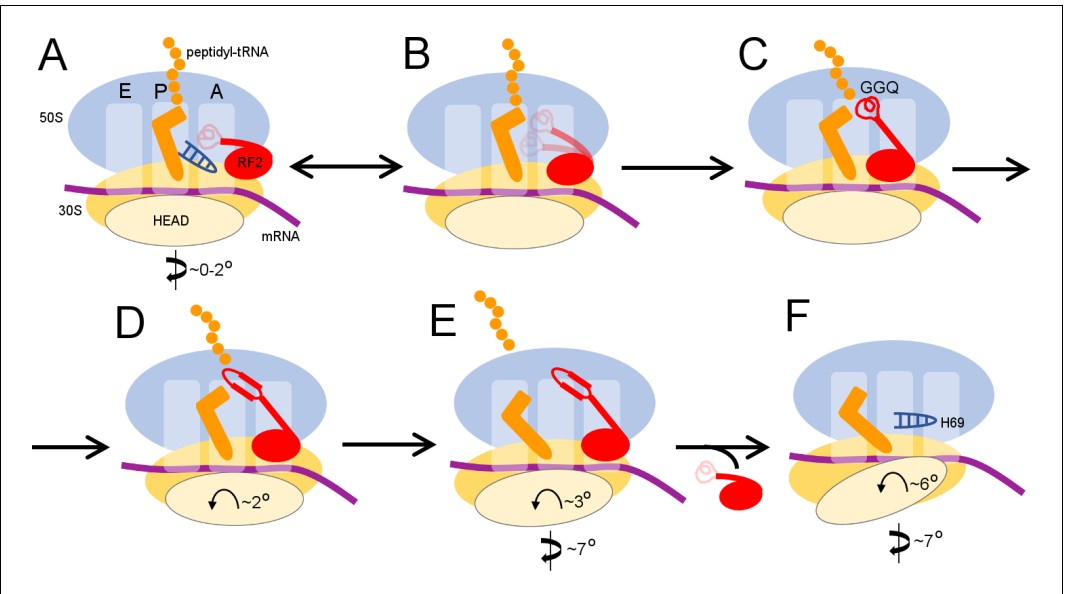

**Figure 4.** Scheme of the termination mechanism.
DOI: https://doi.org/10.7554/eLife.46850.014

In Structure IV, the 30S subunit is rotated 7.5° and the P-tRNA elbow is bound to the L1 stalk, an interaction normally observed with the P/E or E tRNA (*Dunkle et al., 2011*; *Korostelev et al., 2006*; *Selmer et al., 2006*). Unlike P/E tRNA, however, the acceptor arm is oriented between the 50S P and E sites (*Figure 1D*), suggesting that the tRNA adopts a transient state on its way toward the hybrid P/E conformation. This transition coincides with an ~3° rotation (swivel) of the 30S head, characteristic of intermediate stages of tRNA and mRNA translocation (*Abeyrathne et al., 2016*; *Ratje et al., 2010*). The codon-recognition superdomain of RF2 in its canonical conformation moves with the 30S subunit, causing the catalytic domain to shift by ~5 Å from its position in Structure II (*Figure 3A*). Indeed, reduced density for the catalytic domain and domain 1, which bridges the 30S and 50S subunits at the A site periphery (*Figure 1B*), suggests that intersubunit rotation has destabilized RF2. In Structure IV, therefore, RF2 appears to be poised for dissociation and collapsing toward the compact conformation adopted by free RFs (*Figure 3A*) (*JCSG, 2005*; *Shin et al., 2004*; *Vestergaard et al., 2001*; *Zoldák et al., 2007*).

Structure V is incompatible with RF2 binding and provides further insights into RF2 dissociation. The rotated 70S•tRNA structure resembles hybrid P/E states, with the acceptor arm of tRNA directed toward the E site of 50S (*Figure 1E*). However, the central intersubunit bridge B2a—formed by helix 69 (H69, nt. 1906–1924) of 23S rRNA—is disassembled (*Figure 3B–C*). In RF-bound structures (including Structures I to IV, above), the tip of H69 (at nt 1913) locks into the decoding center in a termination-specific arrangement (*Korostelev et al., 2008*; *Korostelev et al., 2010*; *Laurberg et al., 2008*; *Weixlbaumer et al., 2008*). H69 interacts with the switch loop of RFs, where the codon-recognition superdomain connects to catalytic domain 3 (e.g., Structure II, *Figure 3B*). This interaction directs the catalytic domain toward the PTC (*Korostelev et al., 2010*; *Laurberg et al., 2008*) and thus defines the efficiency and accuracy of release factors (*Svidritskiy and Korostelev, 2018a*). In Structure V, however, H69 is disengaged from the decoding center and moved ~15 Å toward the large subunit to pack against H71 near A2602 (nt 1945–1961; *Figure 3C*). The new position of H69 would clash with extended or compact conformations of RF2 (*Figure 3—figure supplement 1*), if the release factor had remained bound to the decoding center.

Release of RF2 also appears to be coupled to rotation (swivel) of the 30S head domain. In RF2-bound structures, the conserved codon-recognition [205]SPF[207] motif (serine-proline-phenylalanine of domain 2) of RF2 interacts with the stop codon and the 30S head near bulged nucleotide C1054 of 16S rRNA. In Structure V, a 6° rotation of the 30S head shifts the position of C1054 by 5 Å from its position in RF2-bound Structure IV, consistent with disassembly of RF2 contacts. The disruption of

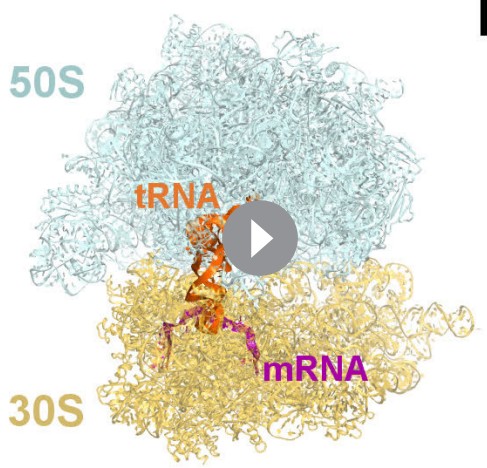

**Animation 1.** An animation showing transitions between the structures of the 70S ribosome upon interaction with RF2.

DOI: https://doi.org/10.7554/eLife.46850.015

decoding-center interactions due to repositioning of H69 and head swivel therefore explain the absence of RF2 in Structure V.

## Mechanism of translation termination by RF2

Our termination structures allow us to reconstruct a dynamic pathway for termination (*Figure 4* and *Animation 1*) and reconcile biophysical and biochemical findings (*Adio et al., 2018*; *Casy et al., 2018*; *Prabhakar et al., 2017*; *Sternberg et al., 2009*). Recent structural studies revealed compact RF conformations on non-rotated 70S ribosomes (*Fu et al., 2018*; *Svidritskiy and Korostelev, 2018a*) that suggest how RFs open at early stages of codon recognition (*He and Green, 2010*; *Hetrick et al., 2009*; *Trappl and Joseph, 2016*). The short-lived transition from compact to open conformation(s) lasts tens of milliseconds (*Fu et al., 2018*), and compact RF2 conformations were not captured in our sample. Compact RF2 was also visualized by cryo-EM on the 70S ribosome in the presence of truncated mRNA and alternative rescue factor A (ArfA) (*Demo et al., 2017*; *James et al., 2016*), suggesting a conserved pathway for initial binding of RFs to non-rotated ribosomes. Together with these recent studies, our work provides an expanded structural view of the termination reaction – from initial codon recognition to peptide release to RF2 dissociation (*Figure 4*).

Termination begins when release factor recognizes a stop codon in the A site of a non-rotated (or slightly rotated) 70S ribosome carrying peptidyl-tRNA (*Figure 4A*). A compact release factor recognizes the stop codon, while its catalytic domain is kept 60 to 70 Å away from the PTC (*Fu et al., 2018*). Separation of the codon-recognition and catalytic functions during initial binding of RF underlies high accuracy of termination (*He and Green, 2010*; *Laurberg et al., 2008*; *Svidritskiy and Korostelev, 2018a*; *Trappl and Joseph, 2016*). Upon binding the decoding center, steric hindrance with P-tRNA forces the RF catalytic domain to undock from domain two and sample the intersubunit space (*Figure 4B*) (discussed in *Svidritskiy and Korostelev, 2018a*). The ribosome remains non-rotated because peptidyl-tRNA bridges the P sites of 30S and 50S subunits. Interactions between the switch loop of RF and the decoding center and H69 direct the catalytic domain into the PTC. The GGQ catalytic loop is placed next to the scissile ester bond of peptidyl-tRNA and stabilized by A2602 (*Jin et al., 2010*; *Laurberg et al., 2008*), so that the Gln backbone amide of the GGQ motif can catalyze peptidyl-tRNA hydrolysis (*Figure 4C*) (*Korostelev et al., 2008*; *Santos et al., 2013*). These ribosome and RF2 rearrangements complete the first stage of termination leading to peptidyl-tRNA hydrolysis—an irreversible step that separates the newly made protein from P-site tRNA and prevents further elongation.

To be recycled, the ribosome must release the nascent protein and RF. The deacylated CCA end of the tRNA exits the 50S P site, which allows movement of A2602. No longer stabilized by the tRNA and A2602, the GGQ loop reorganizes into an extended β-hairpin that blocks the peptide tunnel (Structure II), thus biasing the diffusion of nascent peptide toward the exit of the peptide tunnel at the surface of the 50S subunit (*Figure 4D*). Furthermore, the exit of the CCA end from the P site supports formation of the P/E-tRNA hybrid conformation via spontaneous intersubunit rotation (Structures III and IV), which is also essential for translocation of tRNA-mRNA (reviewed in *Ling and Ermolenko, 2016*; *Noller et al., 2017*). The large counter-clockwise rotation of the small subunit destabilizes the catalytic domain of RF2 (Structure IV), so that it begins to collapse toward domain 2 bound to the 30S subunit (*Figure 4E*). Rotation of the 30S head and detachment of H69 from the decoding center force dissociation of RF2 from the A site of the 30S subunit (Structure V, *Figure 4F*).

Remarkably similar extents of intersubunit rotation and head rotation were observed in recent RF1-bound 70S structures formed with RF3•GDPNP in the presence of apidaecin (which stalls RF1 on the ribosome). In the presence of stalled RF1, however, the intersubunit bridge remains attached to the decoding center (*Graf et al., 2018*). Thus, the ribosome samples globally similar rotation states with RF1 and RF2, with or without RF3. RF3 stabilizes the rotated state (*Ermolenko et al., 2007*; *Gao et al., 2007*; *Jin et al., 2011*; *Zhou et al., 2012*) to promote RF release. However, the role of RF3 in termination is dispensable because intersubunit rotation is an inherent— that is spontaneous and thermally driven—property of the ribosome (*Cornish et al., 2008*). Cold sensitivity of RF3-deletion strains (*Nichols et al., 2011*; *O'Connor, 2015*) perhaps manifests the dependence on RF3 due to perturbation of the intersubunit dynamics at lower temperatures. In the fully rotated state, the post-termination ribosome becomes a substrate for subunit dissociation by EF-G and recycling factor RRF (*Agrawal et al., 2004*; *Dunkle et al., 2011*; *Fu et al., 2016*; *Prabhakar et al., 2017*). Structure V is consistent with structural studies showing ribosome recycling factor (RRF) bound to ribosomes with a dislodged intersubunit bridge B2a (*Borovinskaya et al., 2007*; *Pai et al., 2008*), which helps split the ribosome into subunits. Translation termination and recycling of the release factors and the ribosome therefore rely on the spontaneous ribosome dynamics (*Adio et al., 2018*; *Cornish et al., 2008*; *Ermolenko et al., 2007*; *Prabhakar et al., 2017*; *Sternberg et al., 2009*), coupled with local rearrangements of the universally conserved elements of the peptidyl-transferase and decoding centers.

# Materials and methods

## Key resources table

| Reagent type (species) or resource | Designation | Source or reference | Identifiers | Additional information |
|---|---|---|---|---|
| Strain, strain background (*Escherichia coli*) | *Escherichia coli* MRE600 | (*Cammack and Wade, 1965*) | ATCC 29417 | |
| Strain, strain background (*Escherichia coli*) | *Escherichia coli* BLR (DE3) | Novagen | | |
| Recombinant protein | C-terminally His-tagged K-12 *E. coli* RF2 | (*Demo et al., 2017*) | | |
| Recombinant DNA reagent | *E. coli* RF2 plasmid (vector pET24b) | (*Demo et al., 2017*) | | |
| Chemical compound, drug | *E. coli* tRNA$^{fMet}$ | Chemical Block | | |
| Chemical compound, drug | [$^{35}$S]-methionine | Perkin Elmer | NEG709A500UC | |
| Chemical compound, drug | Biodegradable Scintillation Cocktail | Econo-Safe | | |
| Sequence-based reagent | RNA GGCAAGGAGG UAAAAAUGUGAAAAAAA | IDT | | |
| Software, algorithm | PHENIX | (*Adams et al., 2002*) | https://www.phenix-online.org/ | |
| Software, algorithm | CNS | (*Brunger, 2007*) | http://cns-online.org/v1.2/ | |
| Software, algorithm | PyMOL | (*DeLano, 2002*) | https://pymol.org/2/ | |
| Software, algorithm | Chimera | (*Pettersen et al., 2004*) | https://www.cgl.ucsf.edu/chimera/ | |
| Software, algorithm | Bsoft | (*Heymann and Belnap, 2007*) | https://lsbr.niams.nih.gov/bsoft/ | |
| Software, algorithm | Frealign | (*Lyumkis et al., 2013*) | http://grigorieflab.janelia.org/frealign | |

*Continued on next page*

*Continued*

| Reagent type (species) or resource | Designation | Source or reference | Identifiers | Additional information |
|---|---|---|---|---|
| Software, algorithm | cisTEM | (*Grant et al., 2018*) | https://cistem.org/ | |
| Software, algorithm | MolProbity | (*Chen et al., 2010*) | http://molprobity. biochem.duke.edu/ | |
| Software, algorithm | RSRef | (*Chapman, 1995*; *Korostelev et al., 2002*) | http://www.sb.fsu. edu/~rsref/ | |
| Software, algorithm | SerialEM | (*Mastronarde, 2005*) | http://bio3d.colorado. edu/SerialEM/ | |
| Other | Holey-carbon grids QUANTIFOIL R 2/1, Cu 200 | Quantifoil Micro Tools | | |
| Other | Vitrobot MK4 | FEI | | |
| Other | Titan Krios microscope | FEI | | |
| Other | K2 Summit camera | Gatan | | |

## Preparation of the 70S termination complex with RF2

N-terminally His6-tagged release factor 2 (RF2) from *E. coli* K12 strain was overexpressed in *E. Coli* BLR (DE3) (Novagen) and purified as described for RF1 (*Svidritskiy and Korostelev, 2018a*). Due to overexpression, RF2 is likely incompletely methylated (or unmethylated) at Q252 of the catalytic GGQ motif (*Dinçbas-Renqvist et al., 2000*), which together with the K12-specific T246 residue, renders RF2 catalytically slower (*Dinçbas-Renqvist et al., 2000*), yet functionally and structurally similar to the post-translationally modified release factors (*Korostelev et al., 2008*; *Pierson et al., 2016*; *Weixlbaumer et al., 2008*; *Zeng and Jin, 2018*). *E. coli* tRNA$^{fMet}$ (Chemical Block) was aminoacylated as described (*Lancaster and Noller, 2005*). 70S ribosomes were prepared from *E. coli* (MRE600) as described (*Svidritskiy and Korostelev, 2018a*). Ribosomes were stored in the ribosome-storage buffer (20 mM Tris-HCl, pH 7.0; 100 mM NH$_4$Cl; 12.5 mM MgCl$_2$; 0.5 mM EDTA; 6 mM β-mercaptoethanol) at –80°C. A model mRNA fragment, containing the Shine-Dalgarno sequence and a spacer to position the AUG codon in the P site and the UGA stop codon in the A site (GGC AAG GAG GUA AAA <u>AUG UGA</u> AAAAAA), was synthesized by IDT.

The 70S•mRNA•fMet-tRNA$^{fMet}$ complex with RF2 was assembled in vitro by mixing 400 nM 70S ribosome (all concentrations are specified for the final pre-termination solution) with 12 µM mRNA in buffer containing 20 mM Tris-acetate (pH 7.0), 100 mM NH$_4$OAc, 15 mM Mg(OAc)$_2$. After incubation for two minutes at 37°C, 600 nM fMet-tRNA$^{fMet}$ was added and incubated for five minutes at 37°C. The Mg(OAc)$_2$ concentration was adjusted to 10 mM by adding an equal volume of buffer (20 mM Tris-acetate (pH 7.0), 100 mM NH$_4$OAc, 5 mM Mg(OAc)$_2$) at room temperature. The release reaction was started by adding an equal volume of 4 µM RF2 (in 20 mM Tris, 100 mM NH$_4$OAc, 10 mM Mg(OAc)$_2$) to the pre-termination solution, yielding the following final concentrations: 0.2 µM 70S, 6 µM mRNA, 0.3 µM fMet-tRNA$^{fMet}$, and 2 µM RF2. The solution was incubated for 30 min at room temperature prior to grid preparation and plunging.

The activity of *E. coli* RF2 was tested in an in vitro assay that measures the release of formyl-methionine from the 70S ribosome bound with [$^{35}$S]-fMet-tRNA$^{fMet}$ in the P site, as described (*Svidritskiy and Korostelev, 2018a*). Pre-termination complex was prepared using 0.2 µM *E. coli* 70S ribosomes (all concentrations are for the final reaction mixture upon initiation of the release reaction), 6 µM mRNA and 0.3 µM [$^{35}$S]-fMet-tRNA$^{fMet}$ ([$^{35}$S]-methionine from Perkin Elmer). An aliquot (4.5 µl) of the pre-termination complex was quenched in 30 µl of 0.1 M HCl to represent the zero-time point. Pre-termination complex was split into two tubes. 5 µl of 20 µM RF2 was added to 45 µl of the pre-termination complex in one tube, yielding 2 µM RF2 in the reaction. Buffer without RF2 was added to the second tube to test spontaneous (RF2-independent) hydrolysis of [$^{35}$S]-fMet-tRNA$^{fMet}$. After 6, 15, 90 s, and 6, 15, 30, 60 and 150 min, 5 µl aliquots were quenched in 30 µl of 0.1 M HCl. Samples were extracted with ethylacetate, and the amount of released [$^{35}$S]-labeled N-formyl-methionine was measured using a scintillation counter (Beckman Coulter, Inc) in 3.5 ml of Econo-Safe scintillation cocktail (RPI). All measurements were performed twice (*Figure 1—figure supplement 4*) and time progress data were analyzed and visualized using GraphPad Prizm 7.01.

## Cryo-EM and image processing

Holey-carbon grids (QUANTIFOIL R 2/1, Cu 200, Quantifoil Micro Tools) were coated with carbon and glow discharged with 20 mA with negative polarity for 60 s in a PELCO easiGlow glow discharge unit. 2.8 µl of the 70S•mRNA•fMet-tRNA$^{fMet}$•RF2 complex was applied to the grids. Grids were blotted at blotting force 10 for 4 s at room temperature, 100% humidity, and plunged into liquid ethane using a Vitrobot MK4 (FEI). Grids were stored in liquid nitrogen.

A dataset of 1,069,908 particles was collected on a Titan Krios (FEI) microscope (operating at 300 kV) equipped with a K2 Summit camera system (Gatan), with –0.5 to –1.8 µm defocus. Multi-shot data collection was performed by recording six exposures per hole, using SerialEM (*Mastronarde, 2005*) with beam-image shift. Coma-free alignment was performed using a built-in function. 'Coma vs. Image Shift' from the Calibration menu was used for dynamic beam-tilt compensation, based on image shifts for each exposure. Multi-hole/multi-shot configuration was selected from 'Multiple Record Setup Dialog' to dynamically adjust the beam tilt. Backlash-corrected compensation was applied to each stage movement at the target stage position to reduce mechanical stage drift. 7287 movies were collected. Each exposure (32 frames per movie) was acquired with continuous frame streaming, yielding a total dose of 29.4 e$^-$/Å$^2$. The nominal magnification was 105,000 and the calibrated super-resolution pixel size at the specimen level was 0.667 Å. 6972 movies were selected after discarding those with defects due to ice or image recording. Movies were processed using IMOD (*Kremer et al., 1996*) and binned to pixel size 1.334 Å (termed unbinned or 1 × binned). Movies were motion-corrected, and frame averages were calculated using all 32 frames within each movie (*Figure 1—figure supplement 3*), after multiplying by the corresponding gain reference. cisTEM (*Grant et al., 2018*) was used to determine defocus values for each resulting frame average and for particle picking. The stack and particle parameter files were assembled in cisTEM with the binnings of 1×, 2×, 4×, and 6× (box size of 624 for a non-binned stack).

Data were processed essentially as described in our recent work (*Loveland et al., 2017*) and shown in *Figure 1—figure supplement 1*. FrealignX was used for particle alignment, refinement, and final reconstruction steps, and Frealign v9.11 was used for 3D classification steps (*Lyumkis et al., 2013*). The 6×-binned image stack (1,069,908 particles) was initially aligned to a ribosome reference without release factor and tRNA (PDB 5J4D) (*Svidritskiy et al., 2016*) using 3 cycles of mode 3 (global search) alignment, including data from 20 Å to 300 Å resolution. Subsequently, the 6×-binned stack was refined using mode 1 (3 cycles) to align particles in the 15- to 300 Å resolution range. Using the 4×-binned image stack, the particles were successively refined using mode one by gradually increasing the high-resolution limit to 12- and 10 Å (three cycles for each resolution limit). 3D density reconstruction was obtained using 60% of particles with highest scores. The map contained density for the P-tRNA, mRNA, and RF2.

Data classification was performed using a spherical focused mask with a 45 Å radius, centered roughly between the A-site finger (nucleotides 870–910 of 23S rRNA), protein L11, and peptidyl transferase center. The mask covered most of the ribosomal A site and part of the P site. The particle stack was classified into 16 classes, using the high-resolution limit of 10 Å. The classification resulted in five RF2-containing classes, six classes with tRNA alone (two classes represented Structures I and V), three classes of the 70S ribosome without tRNA, and two classes of the 50S subunit alone. All classes were subsequently reconstructed using the unbinned stack. Three strongest classes with RF2 were merged and extracted from the unbinned stack, and subclassified into 10 classes without masking, using a high-resolution limit of 8 Å. This classification yielded Structure II, four maps that were merged to obtain Structure III, a map with a rotated 70S•RF2 conformation (used for further classification to obtain Structure IV), a lower-resolution map similar to Structure II, and three junk (poorly resolved) classes. The class corresponding to the rotated 70S•RF2 conformation was extracted from the unbinned substack and further classified into three classes using a 25 Å radius spherical-focused mask roughly centered at the elbow of tRNA (high-resolution limit at 12 Å), yielding Structure IV. All final classes were reconstructed in FrealignX (1×-binning) using 95% of particles with highest scores.

The resulting reconstructions varied from 3.3 Å (Structure II) to 4.4 (Structure IV) Å average resolution (Fourier Shell Correlation (FSC) = 0.143). Cryo-EM reconstructions were B-factor sharpened in bfactor.exe (part of Frealign distribution) using different B-factor values (suggested by bfactor.exe) and then used for model building and structure refinements. B-factor from −50 to −100 Å$^2$ was also

used for initial refinements and to visualize higher-resolution details. FSC curves were calculated by FrealignX for even and odd particle half-sets (*Figure 1—figure supplement 2*). FSC between the final models and maps, self and cross-validation FSC (*Figure 1—figure supplement 2*) calculated using Phenix (*Adams et al., 2010*) demonstrate good overall agreement between the structural models and maps, similarly to those in previous work (*Brown et al., 2017*). Blocres (part of Bsoft package v1.9.1; *Heymann and Belnap, 2007*) was used to calculate local-resolution maps (*Figure 1—figure supplement 2*).

## Model building and refinement

Cryo-EM structures of *E. coli* 70S•ArfA•RF2 complex (*Demo et al., 2017*) excluding ArfA, and the 70S•fMet-tRNA^fmet complex with EF-Tu ternary complex (*Loveland et al., 2017*) and improved structure of S2 (*Loveland and Korostelev, 2018*), excluding the ternary complex, were used to create initial models for structure refinement. Chimera (*Pettersen et al., 2004*) was used for fitting the 50S subunit, 30S head, 30S body, mRNA, tRNA, and RF2 domains. Local model fitting was performed in PyMOL (*DeLano, 2002*).

Each structural model was refined by real-space simulated-annealing refinement using atomic electron scattering factors in RSRef, as described (*Svidritskiy et al., 2014*) (*Chapman, 1995*; *Korostelev et al., 2002*). Secondary-structure restraints, comprising hydrogen-bonding restraints for ribosomal proteins and base-pairing restraints for RNA molecules, were employed as described (*Korostelev et al., 2008*). Refinement parameters, such as the relative weighting of stereochemical restraints and experimental energy term, were optimized to produce the stereochemically optimal models that closely agree with the corresponding maps. In the final stage, the structures were refined using Phenix (phenix.real_space_refine) (*Adams et al., 2010*), followed by a round of refinement in RSRef, applying harmonic restraints to preserve protein backbone geometry. B-factors were refined in Phenix. The refined structural models closely agree with the corresponding maps and have good stereochemical parameters, characterized by low deviation from ideal bond lengths and angles, low numbers of protein-backbone outliers and other robust structure-quality statistics, as shown in *Table 2*. Structure quality was validated using MolProbity (*Chen et al., 2010*).

Structure superpositions were performed in PyMOL. For structural comparisons, the distances and angles were calculated in PyMOL and Chimera, respectively. To calculate an angle of the 30S subunit rotation between two 70S structures, the 23S rRNAs were aligned using PyMOL, and the angle between 16S rRNAs body domains (all nucleotide residues excluding 925–1390) was measured in Chimera. To calculate an angle of the 30S-head rotation (swivel) between two 70S structures, the 16S rRNAs of the 30S body were aligned using PyMOL, and the angle between the 16S rRNA residues 925–1390 was measured in Chimera. Ribosome conformations (30S body/subunit rotation and 30S head rotation) for Structures I through V were compared with the crystal structure of the 70S•RF2 complex formed with the UGA stop codon (*Weixlbaumer et al., 2008*).

Unsharpened cryo-EM maps (direct output from Frealign) were deposited in EMDB (Structure I – EMD-20048, Structure II – EMD-20052, Structure III – EMD-20056, Structure IV – EMD-20057, Structure V – EMD-20058). The corresponding PDB models were deposited in RCSB (PDB codes: 6OFX, 6OG7, 6OGF, 6OGG, 6OGI respectively). Figures were prepared in PyMOL.

## Acknowledgements

We thank KangKang Song for help with screening cryo-EM grids and for assistance with data collection at the UMass Medical School cryo-EM facility; Darryl Conte Jr. and members of the Korostelev lab for helpful comments on the manuscript. This study was supported by NIH Grants R01 GM107465 and R35 GM127094.

## Additional information

### Funding

| Funder | Grant reference number | Author |
|---|---|---|
| National Institute of General Medical Sciences | R01 GM107465 | Andrei A Korostelev |

| National Institute of General Medical Sciences | R35 GM127094 | Andrei A Korostelev |

The funders had no role in study design, data collection and interpretation, or the decision to submit the work for publication.

## Author contributions
Egor Svidritskiy, Conceptualization, Data curation, Formal analysis, Validation, Investigation, Visualization, Methodology, Writing—review and editing, Assembled the ribosome complexes, Performed biochemistry, Collected and processed cryo-EM data, Modeled and refined structural models; Gabriel Demo, Data curation, Formal analysis, Investigation, Methodology, Purified ribosome subunits and RF2, Assisted with structure refinements; Anna B Loveland, Formal analysis, Methodology, Assisted with data processing; Chen Xu, Methodology, Implemented multiple-shot (per hole) data acquisition, Optimized data collection; Andrei A Korostelev, Conceptualization, Resources, Formal analysis, Supervision, Funding acquisition, Validation, Investigation, Writing—original draft, Project administration, Writing—review and editing, Supervised the project, Assisted with modeling and structure refinements, Wrote the manuscript with input from co-authors

## Author ORCIDs
Gabriel Demo (iD) https://orcid.org/0000-0002-5472-9249
Anna B Loveland (iD) https://orcid.org/0000-0001-9172-7747
Andrei A Korostelev (iD) https://orcid.org/0000-0003-1588-717X

## Decision letter and Author response
Decision letter https://doi.org/10.7554/eLife.46850.038
Author response https://doi.org/10.7554/eLife.46850.039

# Additional files
## Supplementary files
• Transparent reporting form
DOI: https://doi.org/10.7554/eLife.46850.016

## Data availability
Structural models have been deposited in PDB under the accession codes 6OFX, 6OG7, 6OGF, 6OGG, 6OGI. Cryo-EM data have been deposited to EMDB under the accession codes EMD-20048, EMD-20052, EMD-20056, EMD-20057, EMD-20058.

The following datasets were generated:

| Author(s) | Year | Dataset title | Dataset URL | Database and Identifier |
|---|---|---|---|---|
| Svidritskiy E, Demo G, Loveland AB, Xu C, Korostelev AA | 2019 | Non-rotated ribosome (Structure I) | http://www.rcsb.org/structure/6OFX | Protein Data Bank, 6OFX |
| Svidritskiy E, Demo G, Loveland AB, Xu C, Korostelev AA | 2019 | Non-rotated ribosome (Structure I) | https://www.ebi.ac.uk/pdbe/entry/emdb/EMD-20048 | Electron Microscopy Data Bank, EMD-200 48 |
| Svidritskiy E, Demo G, Loveland AB, Xu C, Korostelev AA | 2019 | 70S termination complex with RF2 bound to the UGA codon. Non-rotated ribosome with RF2 bound (Structure II) | http://www.rcsb.org/structure/6OG7 | Protein Data Bank, 6OG7 |
| Svidritskiy E, Demo G, Loveland AB, Xu C, Korostelev AA | 2019 | 70S termination complex with RF2 bound to the UGA codon. Partially rotated ribosome with RF2 bound (Structure III) | http://www.rcsb.org/structure/6OGF | Protein Data Bank, 6OGF |
| Svidritskiy E, Demo G, Loveland AB, Xu C, Korostelev AA | 2019 | 70S termination complex with the UGA stop codon. Rotated ribosome conformation (Structure | http://www.rcsb.org/structure/6OGI | Protein Data Bank, 6OGI |

| | | | | | |
|---|---|---|---|---|---|
| | | | V) | | |
| Svidritskiy E, Demo G, Loveland AB, Xu C, Korostelev AA | 2019 | 70S termination complex with RF2 bound to the UGA codon. Rotated ribosome with RF2 bound (Structure IV) | http://www.rcsb.org/structure/6OGG | | Protein Data Bank, 6OGG |
| Svidritskiy E, Demo G, Loveland AB, Xu C, Korostelev AA | 2019 | 70S termination complex with RF2 bound to the UGA codon. Non-rotated ribosome with RF2 bound (Structure II) | https://www.ebi.ac.uk/pdbe/entry/emdb/EMD-20052 | | Electron Microscopy Data Bank, EMD-200 52 |
| Svidritskiy E, Demo G, Loveland AB, Xu C, Korostelev AA | 2019 | 70S termination complex with RF2 bound to the UGA codon. Partially rotated ribosome with RF2 bound (Structure III) | https://www.ebi.ac.uk/pdbe/entry/emdb/EMD-20056 | | Electron Microscopy Data Bank, EMD-200 56 |
| Svidritskiy E, Demo G, Loveland AB, Xu C, Korostelev AA | 2019 | 70S termination complex with RF2 bound to the UGA codon. Rotated ribosome with RF2 bound (Structure IV) | https://www.ebi.ac.uk/pdbe/entry/emdb/EMD-20057 | | Electron Microscopy Data Bank, EMD-200 57 |
| Svidritskiy E, Demo G, Loveland AB, Xu C, Korostelev AA | 2019 | 70S termination complex with the UGA stop codon. Rotated ribosome conformation (Structure V) | https://www.ebi.ac.uk/pdbe/entry/emdb/EMD-20058 | | Electron Microscopy Data Bank, EMD-200 58 |

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
