## [Decision Letter]

Thank you for submitting your article "Extensive ribosome and RF2 rearrangements during translation termination" for consideration by *eLife*. Your article has been reviewed by three peer reviewers, including Sjors HW Scheres as the Reviewing Editor and Reviewer #1, and the evaluation has been overseen by John Kuriyan as the Senior Editor. The following individual involved in review of your submission has also agreed to reveal their identity: Israel S Fernández (Reviewer #2).

The reviewers have discussed the reviews with one another and the Reviewing Editor has drafted this decision to help you prepare a revised submission.

Summary:

This paper describes five cryo-EM structures of ribosomal complexes apparently representing different stages of RF2-catalyzed translation termination. The novel observations here are that the tip of domain 3 of RF2 undergoes a rearrangement from an a-helical conformation to a b-hairpin conformation during termination that likely facilitates exit of the newly synthesized protein from the ribosomal polypeptide exit tunnel and that the ribosome can undergo two thermally activated, spontaneous conformational changes, a relative rotation of the ribosomal subunits and a swiveling of the 'head' domain of the small subunit, during termination that likely facilitate dissociation of RF2 from the ribosome. These are interesting observations that significantly extend our understanding of how class I RFs and ribosome conformational changes drive important steps during termination and, as such, all three reviewers recommended publication provided the following comments are addressed adequately.

Essential revisions:

1) The maps provided through the *eLife* system seemed to be unsharpened, as they showed very little detail. However, even after sharpening them with a B-factor of -100A2, they still did not show the expected features for their respective resolutions. My suspicion is that FREALIGN has been used to overfit the data. This should be addressed in the revision. It should be indicated whether gold-standard separation of halves of the data sets were used in the final refinements, or whether those were limited to a specific spatial frequency (like was done in the classifications). If the latter, those frequencies should also be stated in the manuscript, and they should be significantly lower than the claimed resolutions.

In addition: a lot of basic cryo-EM information is missing: the authors should include:

a) At least one micrograph image.

b) Some representative 2D class averages.

c) Local resolution maps of the five structures. Also, because the density of important parts of the maps seems to be a lot worse than the resolution claimed, it would be good to explicitly mention the local resolution of the important features discussed in the main text.

d) For each structure, some zoomed-in figures with the density on top of the molecular model. These figures should be chosen as to validate the resolution claim. For example, in structures I, II and V, the RNA bases should be well separated (they do so at 3.6A), and in structures III and IV β-strands should be well separated, and many (larger) side chains should be visible. In addition, some panels with density for the most important features of each structure should be shown.

e) FSC curves between the refined PDB models and the cryo-EM maps are missing from the manuscript. These should be included. In addition, to evaluate potential overfitting of the models in the maps, for each structure, the authors should also include the FSC curves between a model that was refined in half-map1 versus half-map1, as well as the FSC curve between the same model versus half-map2.

2) There appear to be many self-citations, and there are also a few places where relevant citations are missing or are mis-cited. There are too many to list individually, but, just a few examples: Introduction: the only citation for the phrase "recent biophysical and biochemical findings suggest a highly dynamic series of termination events" is a Rodnina paper. There are many, earlier papers from Ehrenberg, Gonzalez, Puglisi, Green, Joseph, etc. that should be cited here. Introduction: The only citation for the sentence "By contrast, biochemical experiments showed…" is a Green paper. There are earlier papers from Ehrenberg characterizing the effects of the GGQGAQ mutations on the ability of RF3 to accelerate the dissociation of class I RFs from termination complexes that should be cited here. Introduction: There's a sentence that refers to X-ray, cryo-EM, and smFRET studies, but only provides citations to two smFRET studies (Casy et al., 2018 and Sternberg et al., 2009); Introduction: Moazed and Noller, 1989 identified and characterized the P/E hybrid state, but they didn't report that a deacylated P-site tRNA 'samples' the P/E hybrid state 'via a spontaneous intersubunit rotation'--that was later work from Noller and Ha; etc. There are several other instances of missing citations or mis-citations. We would ask that the authors review their citations with an eye for excessive self-citations and for missing citations or mis-citations. In this context, "Ensemble-EM" is also cited as a specific method in the Introduction (Abeyrathne et al., 2016; Loveland et al., 2017). However, this method is more commonly known as (3D) classification of cryo-EM images, and there are many older, more appropriate citations.

3) The sample imaged is a model sample generated by in vitro assembly with purified components of a termination complex. In order to mimic a bona fide termination complex, a short messenger RNA with a strong Shine-Dalgarno sequence followed by a start codon and immediately after by a stop codon was used (mRNA sequence: 5'-GGC AAG GAG GUA AAA AUG UGA AAAAAA-3'). Similar constructs were used to crystallize termination complexes in the past and it has been proven by smFRET experiments that, at least regarding ribosomal inter-subunit dynamics, this model sample behaves similarly to a real termination complex with a peptide linked to the P site tRNA. However, the nature of this model sample should be apparent for the non-specialist reader, highlighting its similarities with a real termination complex but also its possible limitations, especially regarding the "artificial" nature of having a stop codon so close to the Shine-Dalgarno sequence, a situation that never happens in real mRNAs. The authors should explicitly acknowledge this and discuss its implications in the main text.

4) The authors set up a couple of somewhat 'strawman' arguments in claiming that: (i) there are discrepancies in the X-ray, cryo-EM, and smFRET literature with regard to whether ribosomes can undergo intersubunit rotation while bound to class I RFs or whether the non-rotated conformation of the ribosome is stabilized by bound class I RFs and (ii) class I RFs are able to terminate translation and dissociate from the ribosome without the aid of RF3. In the case of (i), it is obviously possible for class I RF-bound ribosomes to undergo intersubunit rotation while still favoring the non-rotated conformation of the ribosome. Moreover, there are enough differences between the cited studies, both in terms of the experimental conditions as well as the technical limitations associated with the various experimental techniques, that it is easy to rationalize differences with regard to whether the class I RF-bound ribosomes would be expected to undergo intersubunit rotation and/or whether the researchers would have been able to capture/observe intersubunit rotation. In the case of (ii), decades of biochemistry from Buckingham, Ehrenberg, Green, and others had already demonstrated that class I RFs are able to terminate translation and dissociate from the ribosome without the aid of RF3, and that the role of RF3 in termination is to accelerate the spontaneous dissociation of the class I RFs. If the authors want to highlight discrepancies in the literature, they should frame them in the context of differences between the studies, experimental design, limitations of the approaches/techniques in the cited papers that might account for such discrepancies. Re-writing this paragraph also in the light of addressing the missing citations and mis-citations pointed out under (2) will further help in toning these arguments down, which would strengthen the manuscript's scholarship.

5) Class I RFs are post-translationally methylated at the Q residue of the GGQ motif of domain 3 and Buckingham, Ehrenberg, and others have shown that this methylation accelerates and/or facilitates class I-catalyzed termination both in vitro and in vivo. Nonetheless, Svidritskiy et al. do not report whether and to what extent their RF2 is methylated. Was RF2 overexpressed in a manner that ensured homogeneous methylation or lack of methylation? If they are overexpressing prfB and not overexpressing prmC, it is likely that they have a mix of methylated and unmethylated RF2. Assuming they are using the wt *E. coli* prfB gene, then the residue at position 246 is a T, rather than an A or S, and Buckingham has shown that, in the wt T246 background, a lack of methylation at Q252 is either seriously detrimental in richer media or lethal in more minimal media. It was felt that a discussion of this issue was not needed in the main text, but that it would be helpful if the authors would include the important/relevant experimental details in the Materials and methods section, for example, did they use the T246 wt *E. coli* variant of RF2; and did they overexpress prmC along with prfB?

6) Structure I is denoted and treated as a pre-termination complex, but that does not seem at all possible given that the sample was prepared by incubating a pre-termination complex for 30 min in the presence of excess RF2, conditions that Figure 1—figure supplement 3 suggest results in robust termination. Structure I is more likely the non-rotated conformation of a post-termination complex that is in equilibrium with its rotated counterpart, Structure V. Based on my reading of the manuscript, it is likely that the authors understand this point, but are nonetheless using this structure as a mimic/analog of a pre-termination complex. If so, I think this is fine, but the authors should explicitly state that this is what they are doing. Related to this, the authors should clarify the description of their activity assay, show the raw data, and/or report 'Released [S35]-fMet (%)' instead of 'Released [S35]-fMet, CPM' on the y-axis of Figure 1—figure supplement 3; as the activity assay is currently described, reported, and plotted, it is impossible to determine whether RF2 is 1% or 99% active in termination.

7) The final sentence of the manuscript reads: "Translation termination and recycling of the release factors and the ribosome therefore rely on the spontaneous ribosome dynamics, triggered by local rearrangements of the universally conserved elements of the peptidyl-transferase and decoding centers". There are a couple of problems with this sentence as written. First, smFRET experiments by Gonzalez, Puglisi, and Rodnina have previously shown that "Translation termination and recycling of the release factors and the ribosome therefore rely on the spontaneous ribosome dynamics" and the relevant articles should therefore be cited here. Moreover, given the data are static structures solved using a sample that is at equilibrium, it is not clear how the authors determined that these spontaneous ribosome dynamics were "triggered by local rearrangements of the universally conserved elements of the peptidyl-transferase and decoding centers". Isn't it equally possible, given the data presented, that the local rearrangements were triggered by the ribosome dynamics?

---

## [Author Response]

Essential revisions:1) The maps provided through the eLife system seemed to be unsharpened, as they showed very little detail. However, even after sharpening them with a B-factor of -100A2, they still did not show the expected features for their respective resolutions. My suspicion is that FREALIGN has been used to overfit the data. This should be addressed in the revision. It should be indicated whether gold-standard separation of halves of the data sets were used in the final refinements, or whether those were limited to a specific spatial frequency (like was done in the classifications). If the latter, those frequencies should also be stated in the manuscript, and they should be significantly lower than the claimed resolutions.

Thank you for pointing out. To exclude high-resolution bias, we have limited the resolution to 12, 10 or 8 Å during different stages of classification, which are significantly lower than the high resolution of the resulting maps (3 to 4 Å). Thus map details that resolve the structure at a substantially higher resolution than 8 Å originate from the data, and cannot be the result of overfitting. We provide this information in Materials and methods. We also specify that the maps have been sharpened by bfactor.exe (part of Frealign distribution) for structural analyses and that unsharpened Frealign maps have been uploaded to EMDB. Also see answers to questions below regarding local resolution and structure/map quality.

In addition: a lot of basic cryo-EM information is missing: the authors should include:a) At least one micrograph image.

A micrograph image is shown in Figure 1—figure supplement 3.

b) Some representative 2D class averages.

We have not performed 2D classification in this work, as unsupervised 3D classification has proven sufficient to obtain ribosome maps from our sample preparations, based on our previous studies.

c) Local resolution maps of the five structures. Also, because the density of important parts of the maps seems to be a lot worse than the resolution claimed, it would be good to explicitly mention the local resolution of the important features discussed in the main text.

We have calculated local-resolution “heat” maps for each map using Blocres and show them in Figure 1—figure supplement 2 (right panels). These maps support the global resolution assignments of the five maps and show well-resolved local features that we discuss, such as peptidyl-transferase center in most resolved structures, such as Structure II.

d) For each structure, some zoomed-in figures with the density on top of the molecular model. These figures should be chosen as to validate the resolution claim. For example, in structures I, II and V, the RNA bases should be well separated (they do so at 3.6A), and in structures III and IV β-strands should be well separated, and many (larger) side chains should be visible. In addition, some panels with density for the most important features of each structure should be shown.

Thank you for this suggestion. Most local features discussed in the manuscript agree reasonably well with local and global resolutions determined at FSC of 0.143 in B-factor sharpened maps, as demonstrated in local views in Figure 2 and Figure 2— figure supplement 2 (demonstrating density views for the detailed structural features we discuss in the manuscript). As suggested by the reviewer, we now also provide views of local density for each structure (Figure 1—figure supplement 2, middle panels), which illustrate local resolution (e.g. separation of RNA bases and secondary structure) consistent with average resolution – for each structure.

e) FSC curves between the refined PDB models and the cryo-EM maps are missing from the manuscript. These should be included. In addition, to evaluate potential overfitting of the models in the maps, for each structure, the authors should also include the FSC curves between a model that was refined in half-map1 versus half-map1, as well as the FSC curve between the same model versus half-map2.

We have added this information in Materials and methods and Figure 1—figure supplement 2 (left panels), for each structure.

2) There appear to be many self-citations, and there are also a few places where relevant citations are missing or are mis-cited. There are too many to list individually, but, just a few examples: Introduction: the only citation for the phrase "recent biophysical and biochemical findings suggest a highly dynamic series of termination events" is a Rodnina paper. There are many, earlier papers from Ehrenberg, Gonzalez, Puglisi, Green, Joseph, etc. that should be cited here. Introduction: The only citation for the sentence "By contrast, biochemical experiments showed…" is a Green paper. There are earlier papers from Ehrenberg characterizing the effects of the GGQGAQ mutations on the ability of RF3 to accelerate the dissociation of class I RFs from termination complexes that should be cited here. Introduction: There's a sentence that refers to X-ray, cryo-EM, and smFRET studies, but only provides citations to two smFRET studies (Casy et al., 2018 and Sternberg et al., 2009); Introduction: Moazed and Noller, 1989 identified and characterized the P/E hybrid state, but they didn't report that a deacylated P-site tRNA 'samples' the P/E hybrid state 'via a spontaneous intersubunit rotation'--that was later work from Noller and Ha; etc. There are several other instances of missing citations or mis-citations. We would ask that the authors review their citations with an eye for excessive self-citations and for missing citations or mis-citations. In this context, "Ensemble-EM" is also cited as a specific method in the Introduction (Abeyrathne et al., 2016; Loveland et al., 2017). However, this method is more commonly known as (3D) classification of cryo-EM images, and there are many older, more appropriate citations.

Thank you. We have updated the references in the manuscript and have included additional references, as suggested, while trying to minimize self-citations.

3) The sample imaged is a model sample generated by in vitro assembly with purified components of a termination complex. In order to mimic a bona fide termination complex, a short messenger RNA with a strong Shine-Dalgarno sequence followed by a start codon and immediately after by a stop codon was used (mRNA sequence: 5'-GGC AAG GAG GUA AAA AUG UGA AAAAAA-3'). Similar constructs were used to crystallize termination complexes in the past and it has been proven by smFRET experiments that, at least regarding ribosomal inter-subunit dynamics, this model sample behaves similarly to a real termination complex with a peptide linked to the P site tRNA. However, the nature of this model sample should be apparent for the non-specialist reader, highlighting its similarities with a real termination complex but also its possible limitations, especially regarding the "artificial" nature of having a stop codon so close to the Shine-Dalgarno sequence, a situation that never happens in real mRNAs. The authors should explicitly acknowledge this and discuss its implications in the main text.

Thank you for raising this important point. We have expanded the introductory section of Results to explain that we use a model complex, which differs from a cellular termination complex containing an entire ORF, yet represents a functional complex as demonstrated by numerous independent studies.

4) The authors set up a couple of somewhat 'strawman' arguments in claiming that: (i) there are discrepancies in the X-ray, cryo-EM, and smFRET literature with regard to whether ribosomes can undergo intersubunit rotation while bound to class I RFs or whether the non-rotated conformation of the ribosome is stabilized by bound class I RFs and (ii) class I RFs are able to terminate translation and dissociate from the ribosome without the aid of RF3. In the case of (i), it is obviously possible for class I RF-bound ribosomes to undergo intersubunit rotation while still favoring the non-rotated conformation of the ribosome. Moreover, there are enough differences between the cited studies, both in terms of the experimental conditions as well as the technical limitations associated with the various experimental techniques, that it is easy to rationalize differences with regard to whether the class I RF-bound ribosomes would be expected to undergo intersubunit rotation and/or whether the researchers would have been able to capture/observe intersubunit rotation. In the case of (ii), decades of biochemistry from Buckingham, Ehrenberg, Green, and others had already demonstrated that class I RFs are able to terminate translation and dissociate from the ribosome without the aid of RF3, and that the role of RF3 in termination is to accelerate the spontaneous dissociation of the class I RFs. If the authors want to highlight discrepancies in the literature, they should frame them in the context of differences between the studies, experimental design, limitations of the approaches/techniques in the cited papers that might account for such discrepancies. Re-writing this paragraph also in the light of addressing the missing citations and mis-citations pointed out under (2) will further help in toning these arguments down, which would strengthen the manuscript's scholarship.

We agree that this paragraph could more clearly state the rationale for this work. Instead of focusing on experimental differences among numerous biochemical and biophysical studies, which may have led to different interpretations, results or other discrepancies, we have rewritten the concluding part of this paragraph to make it clear that our goal was to visualize new 70S*RF2 structures that have been suggested by biophysical studies but have not been structurally characterized.

*5) Class I RFs are post-translationally methylated at the Q residue of the GGQ motif of domain 3 and Buckingham, Ehrenberg, and others have shown that this methylation accelerates and/or facilitates class I-catalyzed termination both* in vitro *and* in vivo*. Nonetheless, Svidritskiy et al. do not report whether and to what extent their RF2 is methylated. Was RF2 overexpressed in a manner that ensured homogeneous methylation or lack of methylation? If they are overexpressing prfB and not overexpressing prmC, it is likely that they have a mix of methylated and unmethylated RF2. Assuming they are using the wt E. coli prfB gene, then the residue at position 246 is a T, rather than an A or S, and Buckingham has shown that, in the wt T246 background, a lack of methylation at Q252 is either seriously detrimental in richer media or lethal in more minimal media. It was felt that a discussion of this issue was not needed in the main text, but that it would be helpful if the authors would include the important/relevant experimental details in the Materials and methods section, for example, did they use the T246 wt E. coli variant of RF2; and did they overexpress prmC along with prfB?*

We now provide this additional information in the experimental section.

6) Structure I is denoted and treated as a pre-termination complex, but that does not seem at all possible given that the sample was prepared by incubating a pre-termination complex for 30 min in the presence of excess RF2, conditions that Figure 1—figure supplement 3 suggest results in robust termination. Structure I is more likely the non-rotated conformation of a post-termination complex that is in equilibrium with its rotated counterpart, Structure V. Based on my reading of the manuscript, it is likely that the authors understand this point, but are nonetheless using this structure as a mimic/analog of a pre-termination complex. If so, I think this is fine, but the authors should explicitly state that this is what they are doing. Related to this, the authors should clarify the description of their activity assay, show the raw data, and/or report 'Released [S35]-fMet (%)' instead of 'Released [S35]-fMet, CPM' on the y-axis of Figure 1—figure supplement 3; as the activity assay is currently described, reported, and plotted, it is impossible to determine whether RF2 is 1% or 99% active in termination.

We agree and we have added a clarification that Structure I is likely deacylated but the ribosome and tRNA are similar to pre-reaction structures. We have also updated the supplementary figure to show the time progress curves with fractions of released fMet instead of CPM (now Figure 1—figure supplement 4).

7) The final sentence of the manuscript reads: "Translation termination and recycling of the release factors and the ribosome therefore rely on the spontaneous ribosome dynamics, triggered by local rearrangements of the universally conserved elements of the peptidyl-transferase and decoding centers". There are a couple of problems with this sentence as written. First, smFRET experiments by Gonzalez, Puglisi, and Rodnina have previously shown that "Translation termination and recycling of the release factors and the ribosome therefore rely on the spontaneous ribosome dynamics" and the relevant articles should therefore be cited here. Moreover, given the data are static structures solved using a sample that is at equilibrium, it is not clear how the authors determined that these spontaneous ribosome dynamics were "triggered by local rearrangements of the universally conserved elements of the peptidyl-transferase and decoding centers". Isn't it equally possible, given the data presented, that the local rearrangements were triggered by the ribosome dynamics?

We agree and we have added the references to this section of the manuscript. We have substituted “triggered by” with “coupled with”.